# Insights into Current Education of Acupuncture as a Non-Conventional Therapy in Portugal

**DOI:** 10.3390/healthcare11101389

**Published:** 2023-05-11

**Authors:** Xiao Ye, Lara Lopes, Ricardo Teixeira, Ying Wang, Jorge Pereira Machado

**Affiliations:** 1IZCMC—Institute of Zhejiang Chinese Medical Culture, Zhejiang Chinese Medical University and Key Cultivating Base of Zhejiang Philosophy and Social Sciences, Hangzhou 310053, China; yexiao@zcmu.edu.cn; 2HMS—Humanities and Management School, Zhejiang Chinese Medical University, Hangzhou 310053, China; 3SMH—School of Medical Humanities, Nanjing University of Chinese Medicine, Nanjing 210023, China; 4ICBAS—School of Medicine and Biomedical Sciences, University of Porto, 4050-313 Porto, Portugal or llpsique@gmail.com (L.L.); jmachado@icbas.up.pt (J.P.M.); 5CBSin—Center of BioSciences in Integrative Health, 4250-105 Porto, Portugal; 6ETCMA—European Traditional Chinese Medicine Association, 3811 LS Amersfoort, The Netherlands; ricardo.teixeira@essnortecvp.pt; 7SBMS—School of Basic Medical Sciences, Zhejiang Chinese Medical University, Hangzhou 310053, China

**Keywords:** acupuncture, healthcare education, healthcare legislation, non-conventional therapies, complementary and alternative medicine, CAM, Portugal, European Union

## Abstract

Acupuncture, as an ancient practice for healthcare in China, is now widely used in the world and regarded as a non-conventional therapy (NCT) in many Western countries. In Portugal, acupuncture has been structured and well regulated for the market of teaching and clinical practice, but little effort has been put in to explore it in depth. This article aims to disclose the current education of acupuncture as a NCT in Portugal through investigation of acupuncture laws, field surveys, teaching work, and interviews with people from the NCT field. We found that according to the academic norms and rules of education in Portugal, there is a gradual difficulty in the progression and maintenance of the degree training dynamics. The reasons are the lack of more tolerant transitional measures and many practical difficulties confronted by the institutions that embark on these complementary programs. Therefore, it will be necessary to promote additional programs and measures to avoid a total emptiness of the teaching of acupuncture and at the same time losses of clinicians, competencies, and quality of information that are difficult to recover. It could be very meaningful and thought provoking to the future development and improvement of acupuncture in Portugal and in other countries that welcome acupuncture and intend to have better legislation and application.

## 1. Introduction

Acupuncture is a technique in which practitioners insert fine needles into the skin to treat health problems. It originated from traditional Chinese medicine at least 2500 years ago but has gained popularity worldwide since the 1970s [1]. According to the WHO global report on traditional and complementary medicine 2019, acupuncture remains the most common form of traditional medicine practice among its 113 Member States out of all 194 [2]. The WHO recognizes 28 diseases, symptoms, or conditions for which acupuncture has been shown as an effective form of treatment, and it also recognizes acupuncture’s therapeutic effects for over 55 diseases, symptoms, or conditions [3]. A recent overview of acupuncture systematic reviews found that of 77 diseases investigated, acupuncture showed a moderate or large effect with moderate or high certainty evidence in eight diseases or conditions, namely post-stroke aphasia, neck and shoulder pain, myofascial pain, fibromyalgia-related pain, non-specific lower back pain, lactation success rate within 24 h of delivery, vascular dementia symptoms, and allergic rhinitis nasal symptoms [4].

In the West, acupuncture as a distinct therapeutic system is recognised by law in 12 European Union (EU) Member States, and the medical association/council/chamber in many countries has recognised acupuncture as an additional medical qualification [5]. Portugal not only has recognized medical acupuncture practiced by medical doctors but also is the first EU country to pay attention to the protection of the acupuncturist title and provide specific legislation on acupuncture as a non-conventional therapy (NCT) at the national level [6]. The majority of Portuguese medical doctors recently considered that the competence in medical acupuncture of the Portuguese Medical Association (Ordem dos Médicos) should remain available [7]. Since the situation of medical acupuncture for medical doctors remains generally the same in these years and is similar to that in other European countries, this article will mainly focus on non-doctor acupuncture as a representative of non-conventional therapies with great shifts and distinct features in Portugal.

Acupuncture as a non-conventional therapy in Portugal was legally regulated together with homeopathy, osteopathy, naturopathy, phytotherapy, and chiropractic in Law 45/2003 approved by the Assembly of the Republic, on 15 July 2003 and enacted by President Jorge Sampaio and Prime Minister José Barroso on 8 August 2003 [8]. It offered the first basic legal framework for according to non-conventional therapies in Portugal and initiated a series of more specific acupuncture laws thereafter. The acupuncture profession in Portugal has thus developed with new trends in recent years and it is the same case of acupuncture education, which is essential to its professional development, lays the foundation for the safe and best practice of acupuncture, and supports the sustainable development of acupuncture in the country. However, little effort has been put in to explore the specific current situation of acupuncture education in Portugal, which is informative, diverse, and flawed in some respects. Therefore, the aim of this study was to determine Through investigation of acupuncture laws, field surveys, occasional teaching work, and interviews with people related to the field in Portugal, we would like to share our vision of the current acupuncture education in Portugal and expose problems regarding the legislation, teaching contents, staff, research, internship, etc. It is then possible to provide constructive suggestions, which could be very meaningful and thought-provoking to the future development of acupuncture in Portugal, in Portuguese-speaking countries and in other countries that welcome acupuncture and intend to have better legislation and application, even for other NCTs.

## 2. Methodology

The methodology selected for this short communication was the narrative review.

The authors conducted it through investigation of acupuncture laws, field surveys, occasional teaching work, and interviews with open questions to people related to the field in Portugal.

We searched at Diary of the Republic of Portugal (DR) and through the Central Administration of Health System (ACSS) website for laws related to non-conventional therapies and Acupuncture.

Moreover, we recruited 19 people for field surveys. After duplicates removal, we divided interviewees in two groups according to the specificity of our questions (Figure 1). There were four questions made to all the people interviewed (Table 1) and seven more asked only to institution-affected individuals (Table 2).

## 3. Results

As for the education of acupuncture as a non-conventional therapy in Portugal, the policies and institutions are two essential aspects that determine its development, tendency, and quality. The former provides a framework, while the latter is the main body to put policies into practice.

### 3.1. Major Policies

After the initial legislation of acupuncture in 2003, there was no specific law concerning acupuncture education until 2013 when Law 71/2013 was enacted, which regulated acupuncture professional activity and application, and determined access to the acupuncture profession with the prerequisite of acquiring higher educational training [9].

On 12 September 2014, a scoring system for the transitional period, a time when there was no degree education in acupuncture in Portugal, was established to help determine applicant availability for a license to exercise acupuncture in Portugal, including education level, professional and complementary training, internships, and research [10].

On 8 October 2014, acupuncture as a non-conventional therapy was defined in Ordinance 207-F/2014, stating that “Acupuncture is a therapy that uses methods of diagnosis, prescription and treatment based on axioms and theories of acupuncture meridians, points and reflex zones of the human body to prevent and treat energetic, physical and psychic disharmonies”. It also clearly states that the theory and practice of acupuncture are based on traditional Chinese medicine (TCM) [11].

On 5 June 2015, the curricular content and access requirements of acupuncture undergraduate education were specified under the Law 172-C/2015, which indicated full legislation of acupuncture in Portugal. Specifically, this Law defined the content of higher education degree of acupuncture to the undergraduate level being carried out by polytechnic institutes, which are polytechnic higher education schools that may or may not be integrated into universities. Applicants for the acupuncture degree education must take the college entrance exams that integrate the areas of biology and physics/chemistry. Its curricular content requires 240 ECTS (ECTS is an abbreviation of European Credit Transfer and Accumulation System and 1 ECT equals to about 27 h of training) in eight semesters and includes basic and clinical subjects of modern medicine, basic theories of TCM, and practice of acupuncture, internship, and some other subjects, such as communication sciences, initiation to research in acupuncture, ethics and deontology, and legislation. There are 130 ECTS for contents of the traditional theories and practice of acupuncture, taking up 54% of the total curricular content [12].

On 9 September 2019, supplementary Law 109/2019 setup a deadline for candidates without professional higher education and degree and stated that they can still apply for the license until 31 December 2025 or until the first bachelor’s degree is granted in each of the regulated non-conventional therapies [13].

### 3.2. Major Education Institutions

As early as in the 1980s, one non-conventional therapies school in Portugal had been founded by Araújo Ferreira and people could learn “naturopathy, homeopathy, and acupuncture” [14]. Later, more similar schools were established, and many relevant associations and some universities offered courses in acupuncture. With the requirements for acupuncture degree education being released by the government in 2015, institutes providing acupuncture degree education have emerged these years.

Various institutions in Portugal provide education on acupuncture as a non- conventional therapy and they can be generally classified into three categories according to the teaching purpose, content, and the nature of the institutions, namely degree education in polytechnic institutes, non-degree education in classic non-superior schools, and other institutes.

#### 3.2.1. Acupuncture Degree Education in Polytechnic Institutes

In 2017, the first licentiate course in acupuncture for a bachelor’s degree accredited by the Agency for Assessment and Accreditation of Higher Education (A3ES) in Portugal started to open in a public institution of higher education, namely the Escola Superior de Saúde, Instituto Politécnico de Setúbal [15]. It enrolled students for four years with 12 to 20 attendees in each session. However, in January 2021, it was discredited by the A3ES because it had only 30% of professors with doctorates or specialized in the acupuncture study cycle. This contributed to the insufficiency of full-time professors in the fundamental area of the study cycle as well as the low scientific production in acupuncture. Additionally, the faculty was in great instability and the institute did not present a concrete plan or a clear institutional commitment to meet this legal requirement in the short term [16].

Nevertheless, three other polytechnic institutes are still officially recognized to offer bachelor’s degree education in acupuncture, namely the Escola Superior de Saúde Norte da Cruz Vermelha Portuguesa, accredited on 12 June 2018 [17], the Instituto Politécnico da Lusofonia-Escola Superior de Saúde Ribeiro Sanches, accredited on 21 June 2018 [18], and the Instituto Piaget-Escola Superior de Saúde Jean Piaget de Vila Nova de Gaia, accredited on 19 July 2018 [19], respectively. More available information is shown in Table 3. Although they also have similar difficulties to those of the Instituto Politécnico de Setúbal, they are active and stumbling forward.

#### 3.2.2. Acupuncture Non-Degree Education in Classic Non-Superior Schools

Portugal’s acupuncture education in classic non-superior schools has a relatively long history. Most of them were established in the 1990s, under non-official and non-academic connections with the Ministry of Science, Technology, and Higher Education of Portugal. However, they usually have sufficient teaching staff, a well-organized teaching system, and high-quality education. In fact, many of them have close connections with Chinese Medicine universities in China and have Chinese teaching staff or internships in China. Before 2017, they were the only institutes in Portugal that could provide four to five years of acupuncture education and had a few flourishing years with hundreds of students.

Nevertheless, since the first bachelor’s degree in acupuncture had been granted in 2021 by the polytechnic institute, and according to article 19 of Law 109/2019 that has been illustrated above, those who graduated from non-superior schools without an acupuncture degree have been unable to apply for acupuncture license after August 2021. This means that people learning acupuncture who graduated from these non-superior schools will be unable to obtain an acupuncture license anymore.

Hence, these non-superior schools are facing a major challenge and the shortage of students is evident. If they fail to adapt to the legal regime of higher education institutions in the next years, their future development would be gloomy. A serious case in point is the Instituto de Medicina Tradicional (IMT), an institute created in 1997 that provided training in the areas of unconventional therapies (including acupuncture), which once had establishments in Lisbon, Braga, and Porto. It also had a close relationship with many universities of Chinese Medicine in China, such as Jiangxi University of Chinese Medicine, Zhejiang Chinese Medical University, and Chengdu University of Traditional Chinese Medicine. Unfortunately, it closed its activity without warning in April 2021 and left hundreds of students in the whole country without any support [20].

Similarly, the legislation is also a blow to other non-superior schools, which reduced their teaching programs of acupuncture and TCM greatly. These non-superior schools currently continue to provide TCM teaching programs with acupuncture courses being integrated as well as some very specific and restricted continuing training courses in acupuncture. The most well-known ones in Portugal are the UMC, the Escola de Medicina Tradicional Chinesa (ESMTC), and the Instituto Português de Naturologia (IPN). Some of their information is shown in Table 4.

#### 3.2.3. Acupuncture Education in Other Institutions

Besides the above-mentioned polytechnic institutes and non-superior schools, some other institutions, such as Instituto Van Nghi (IVN), Instituto Medicina Integrativa, Atlântico Business School, Colégio Português de Medicina Natural (CPMN), Confucius Institutes in Portugal, and Sino-Portugal TCM Center, offer some courses, seminars, and lectures on acupuncture. Especially, the IVN supplies a wide range of courses of acupuncture and TCM; the Confucius Institute of University of Coimbra with Zhejiang Chinese Medical University as one of the partners not only has an optional curricular unit (2 ECTS) containing acupuncture periods for medical students, but also holds seminars and delivers lectures on acupuncture; and the Atlântico Business School provides post-graduate education of TCM, with acupuncture courses included and scientific research as a mandatory milestone to conclude the course.

These institutions have enriched the education of acupuncture as a non-conventional therapy in Portugal and helped people of different walks to learn and understand acupuncture based on traditional Chinese medicine better.

### 3.3. Medical Acupuncture in Portuguese Universities

Although the legislation of acupuncture tends to regulate different ways of acupuncture practice under one profession, medical doctors insist on listing medical acupuncture as a special branch. Hence, to put the education of acupuncture as a non-conventional therapy in a bigger context, it is necessary to have a brief introduction to the education of medical acupuncture, which is exclusively learned and practiced by medical students and doctors.

In the mid-1980s, the first course on acupuncture exclusively for medical doctors, taught by Portuguese teachers, ran for a couple of years [21]. On 19 August 2001, the Portuguese Medical Society of Acupuncture (Sociedade Portuguesa Médica de Acupunctura, SPMA) was made official, representing Portuguese doctors who practice acupuncture. It aims to contribute to the promotion, dissemination, and scientific research of this therapeutic modality. In May 2002, the Portuguese Medical Association approved the Competency in Medical Acupuncture (Competência em Acupunctura Médica) to regulate the scope of practice and define the skills that a medical doctor should have to be allowed to practice acupuncture. This training must be carried out in Universities approved by the Ministry of Science, Technology, and Higher Education, through an appropriate curriculum. Many universities provide postgraduate courses in acupuncture, such as the Faculty of Medicine at the University of Porto (FMUP) and ICBAS—School of Medicine and Biomedical Sciences at the University of Porto, the Faculty of Medicine at the university of Coimbra, the Faculty of Medical Sciences at the New University of Lisbon, and the School of Medical Sciences at the University of Minho [22,23]. Hundreds of doctors have been able to offer acupuncture as a medical treatment to their patients.

However, the contents and training duration of such a post-graduate program are different among these universities. Especially, there is a trend that the content of basic theories of TCM has gradually been ignored or cancelled by more recently established programs. Table 5 is a list of the acupuncture training plans among these four universities.

Some Portuguese researchers found that teaching acupuncture to medical doctors is best achieved by teaching Western style acupuncture, focusing on the needling technique, and basing the treatment plans on sound knowledge of anatomy and physiology [17]. Certainly, medical acupuncture is much easier for medical doctors to comprehend, study, and practice. Thus, it is becoming more popular in acupuncture education among universities.

## 4. Discussion

The comprehensive and successful legislation on acupuncture as a non-conventional therapy in Portugal is encouraging. Students with a bachelor’s degree in acupuncture would surely be more welcomed in Portuguese-speaking and in European countries, and people from other countries can come to Portugal to obtain a bachelor’s degree in acupuncture. The quality and safety of acupuncture practice are also guaranteed to some extent. Laws are important to legally frame and protect the public and the professionals, but they need to be considered in a broader context (i.e., educational and health systems) to ensure sustainability [24]. However, a series of problems concerning the education of acupuncture as a non-conventional therapy in Portugal are still pending resolution.

### 4.1. Lack of Full-Time Professors

Lacking full-time professors for acupuncture degree education is a big problem, and fundamentally, it is because the schools do not have enough funds to contract them. According to Article 6 (Attribution of the Bachelor’s Degree) of the Decree-Law 63/2016, the faculty for a bachelor’s degree education is considered to be: (a) appropriate when the total faculty is constituted by a minimum of 60% of professors on a full-time basis; (b) academically qualified when a minimum of 15% of professors on a Ph.D. degree; (c) specialized when a minimum of 50% of the faculty are specialists or professors with a doctoral degree with recognized professional experience and competence in the area or areas of fundamental training of the study cycle [25]. However, for institutions of acupuncture degree education it is difficult to meet such requirements because contracting so many full-time professors and specialists would be a huge investment, while their income from this degree education is very low at the initial stage. This problem could possibly relieve a little if these schools can cooperate with some universities in China, who could provide acupuncture professors with high quality and low or even no cost to them. Portugal and Chinese governments signed an Agreement on the Recognition of Academic Degrees and Periods of Higher Education Studies on 12 January 2005. It was approved by the Portugal Council of Ministers on 2 May 2013 [26]. There are a vast number of acupuncture teachers with doctoral degree in China, and many of them would be willing to work in European countries, including Portugal. Moreover, many Portuguese and Europeans have obtained doctoral degrees in acupuncture from universities in China. Therefore, relevant people and organizations need to promote the enforcement of the decree of mutual recognition of academic degrees with China in the field of acupuncture and non-conventional therapies. Once it can be put into practice, the acupuncture teaching staff would be reinforced, and the acupuncture degree education could be more sustainable. In addition, an alternative way could be like in Australia where acupuncture or TCM professors from China or other countries with doctoral degrees can be temporarily recognized when they are teaching and working in these polytechnic institutes.

### 4.2. Lack of Scientific Research

Doing research is important for the advancement of a discipline and the prerequisite for establishing all the bachelor, master, and doctoral degree educational programs. However, few scientific research studies have been conducted in the field of acupuncture by institutions in Portugal. The authors made a general search in the Pubmed with <acupuncture> or <acupuntura> and <Portugal> as the search terms in the recent 10 years (2012–2021), and no more than 50 papers were found. Surprisingly, neither the non-superior schools that provide the non-degree education of acupuncture, nor the polytechnic institutes that offer acupuncture degree education, are in the list of any SCI-indexed papers as the author unit. Most of the published acupuncture research papers were contributed by the University of Porto. Surely, it is understandable that the polytechnic institutes offering acupuncture degree education are just at the initial stage. They do not have enough research facilities, much research may not be reflected by published papers, and many teachers conduct research in research centres or institutes that are not affiliated to these polytechnic institutes. We would just like to point out that acupuncture education institutes in Portugal need to attach great importance to the research issue for future development, and perhaps the first step is to cooperate with some universities that have great strength in acupuncture research both at home and abroad.

### 4.3. No Master and Doctoral Degree Education

Though there has been bachelor’s degree education in acupuncture provided by some polytechnic institutes, no master and doctoral degree education programs in acupuncture are in existence. High-quality development of acupuncture requires practitioners with master’s and doctoral degrees in acupuncture, so it has been necessary to initially specify some regulations on the contents according to education and then some institutes may gradually be able to meet the requirements. However, surely, there is a wide gap even in the basic requirement of the master’s degree education, because according to article 16, Chapter 3, Decree-Law 63/2016, the faculty of master’s degree education is considered to be (a) appropriate when the total faculty is constituted by a minimum of 75% of full-time faculty; (b) academically qualified when a minimum of 60% of teachers have the doctoral degree; (c) specialized when: (i) a minimum of 50% of the total teaching staff are specialists with recognized experience and professional competence in the area or areas of training fundamentals of the study cycle or by doctors specialized in that area or areas; (ii) a minimum of 40% of the total faculty are doctorates specialized in the area or areas of fundamental training of the study cycle [21]. The demand of the teaching staff is much higher than that for the bachelor’s degree, so it is still far from reaching and patience is required, but relevant people need to be well prepared for future development.

### 4.4. Limited Student Vacancies

Among all the polytechnic institutes that can provide acupuncture degree education, the maximum number of vacancies each year for studying acupuncture is 30 and the entry condition is restricted to a high school science background, which has a serious impact on source potential acupuncture learners because most high school graduates with a background in science are reluctant to choose TCM (including acupuncture) as a target for their studies since TCM is used as an alternative medical treatment in the West [27]. What is more, although professions such as physiotherapy and nursing need students to have a high school science background, they only require students to have the score from a biology exam, while acupuncture degree education requires the scores both of biology and chemistry/physics exams. In addition, these polytechnic institutes are all private, which have difficulties receiving funds from the government and the tuition fee for students is much higher. This means that their finances are very limited to carry out teaching and research activities, provide vacancies for internships, and maintain a stable group of working staff and specialists. A possible solution could be to expand the maximum number of vacancies, allow high school graduates just with the score of biology exam to apply for the acupuncture degree education, and establish a bachelor’s degree education in TCM with which many teachers and working staff can be shared. The Portugal government should also be aware of the difficulties and initial stage of the degree education of acupuncture and other non-conventional therapies and provide moderate and flexible policies. Otherwise, they would probably follow the suit of Escola Superior de Saúde, Instituto Politécnico de Setúbal, which had stopped enrolment of acupuncture students in degree education in 2021, being unable to move forward smoothly.

### 4.5. Limited Internship Clinics

In big cities such as Porto and Lisbon, there are many acupuncture clinics owned by professional Chinese or Portuguese acupuncturists. If the polytechnic institutes providing acupuncture degree education are in these cities, the acupuncture internship would not be a big problem, such as the Instituto Politécnico da Lusofonia. However, many of the polytechnic institutes are in small cities and far away from big cities, where there are very few acupuncture clinics, and it is unfair for the patients and clinics with many students crowding in to have observations, practices, and discussions. Moreover, since the budget of these institutes is tight, they are unable to afford the lodging and transport fee for students to travel a long journey for internship, which also has safety concerns. The best way may be probably for such institutes to set up some local teaching clinics for acupuncture, which would offer cheaper and even free treatments for patients at the beginning. With appropriate advertisement and evident efficacy, it not only enhances the reputation of the institute and promotes the popularity of acupuncture, but also solves the problem of acupuncture internship for students. Furthermore, internship cooperation programs could be established with TCM universities in China, where students can have more opportunities to get in touch with patients and learn different ways of practice.

### 4.6. A Gap between Traditional Acupuncture and Medical Acupuncture

Both the polytechnic institutes and non-superior schools provide acupuncture education based on traditional theories (traditional acupuncture), while the medical faculties in the university usually offer medical acupuncture more based on biomedicine. Since there is no doctoral degree education in acupuncture in Portugal and acupuncture professors with a doctoral degree are urgently needed in the polytechnic institutes, such professors would undoubtedly come from universities, who would study medical acupuncture rather than traditional acupuncture as a non-conventional therapy. Even if there would be more acupuncture professors with doctoral degree from universities in the polytechnic institutes for acupuncture degree education, the teaching quality of acupuncture traditional theories would be influenced, and the future development of acupuncture as a non-conventional therapy could be surely affected. It is important that acupuncture based on traditional theories of Chinese medicine and medical acupuncture should not separate from each other as both are under the profession of acupuncture and have their own strengths and shortcomings. Resultantly, a closer connection between the polytechnic institutes and universities in acupuncture education should be established. Both the traditional and medical acupuncture may need to be lectured and known by students.

### 4.7. The Losing of Classic Non-Superior Schools

Although the classic non-superior schools providing education on acupuncture and other non-conventional therapies contributed greatly to the cultivation of a massive number of qualified practitioners and to the successful legislation, they are now in a very awkward time and will probably vanish because students who graduated from them do not have a bachelor’s degree and they have been unable to be qualified applicants for the license for practicing acupuncture and will be disqualified from the TCM license after 31 December 2025. Such non-superior schools must work hard to transfer into polytechnic institutes or cooperate with polytechnic institutes, and the government should help them as much as possible. Law 109/2019 orders that these non-superior schools providing non-conventional therapies’ education must make the legal transition to higher education schools by 31 December 2023 [13], but it seems none of them will be successful as the deadline is just one year away. It would be a tremendous loss to acupuncture education and other non-conventional therapies in Portugal if all of them come to bankruptcy soon.

### 4.8. Lack of Reference Books Translated from Chinese

Language is still a barrier between having a better understanding of Eastern philosophies and theories of acupuncture. Though there have been some acupuncture books in Portuguese (mostly Brazilian Portuguese), they are not in full range and the quality cannot be guaranteed. Few specialists and linguists are involved in the Portuguese translation of acupuncture and related books, and people still need to refer to professional books in English as the main source. Since acupuncture originated in China and there are numerous related books in China waiting to be translated, scholars in both China and Portugal need to work together to apply for translation research programs to translate acupuncture books and ancient classics into Portuguese with high quality. However, great attention needs to be paid to cultural differences between the East and West in the translation, because some concepts which may be easily understood by the Chinese could be misleading for the West; and some therapies which may be taken for granted in China could be illegal in the West.

## 5. Limitations and Future Perspectives

Due to the limitations of time, funds, and communication, only a minority of institutions providing acupuncture education had been visited and a few relevant people have been interviewed. The data collected would thus not be comprehensive enough, and some underlining problems might not have been revealed. More field work and interviews could be further carried out to discover a more detailed picture of acupuncture education in Portugal.

## 6. Conclusions

Acupuncture education in Portugal has been diverse and widespread in universities, polytechnic institutes, non-superior schools, and other institutions in different forms. The legislation of acupuncture as a non-conventional therapy has been comprehensive and successful, which stipulates the requirement of a bachelor’s degree for applying for the professional license and specifies the requirements and contents of the degree education. This guarantees the quality of acupuncture education and safe practice in Portugal to some extent, and it is the pioneer in the Portuguese-speaking countries and Europe, though there are still many problems waiting to be resolved and urgent actions are required to carry out better and sustainable future development.

## Figures and Tables

**Figure 1 healthcare-11-01389-f001:**
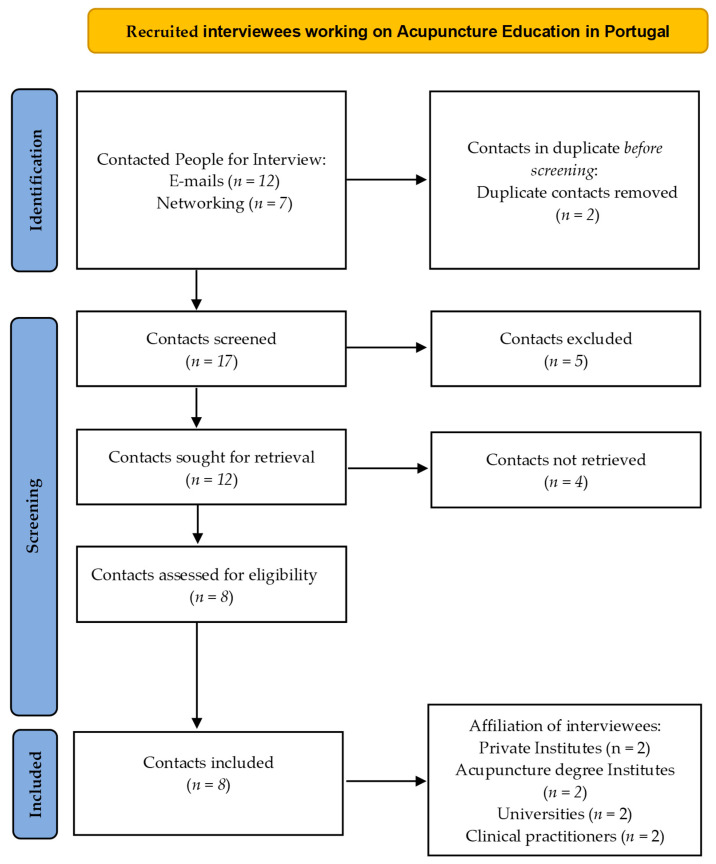
Sample of Interviewees flowchart.

**Table 1 healthcare-11-01389-t001:** Questions made to all the interviewees.

General Questionnaire
Q1: Which was the first non-conventional therapies school in Portugal?
Q2: What type of institutions providing Acupuncture education in Portugal do you know?
Q3: What is your opinion regarding medical Acupuncture teaching?
Q4: In your opinion, what are the main problems of Acupuncture education in Portugal?

**Table 2 healthcare-11-01389-t002:** Questions for specific interviewees according to their affiliation.

Individual Questionnaire
Qa: How many students did you enroll per year at Acupuncture course?
Qb: What strengths and difficulties did you consider that your Institution might have?
Qc: Did your Institution have any cooperation with Universities in China?
Qd: If you answer yes at Qc, would you name them?
Qe: How many teachers do you have for acupuncture degree education?
Qf: How many of them are full/part time?
Qg: How much is the annual fee for degree education?

**Table 3 healthcare-11-01389-t003:** Acupuncture Degree Education in Polytechnic Schools.

Polytechnic School	Duration	Vacancies Each Year	Teachers	ECTs	Annual Fee
Escola Superior de Saúde Norte da Cruz Vermelha Portuguesa	4 years/ 8 semesters	30	Full time: 7 Part time: 24	240	3420.00 €
Instituto Politécnico da Lusofonia-Escola Superior de Saúde Ribeiro Sanches	4 years/ 8 semesters	30	Unknown	240	4731.60 €
Instituto Piaget-Escola Superior de Saúde Jean Piaget de Vila Nova de Gaia	4 years/ 8 semesters	25	Full time: 2 Part time: 16	240	4350.00 €

**Table 4 healthcare-11-01389-t004:** Acupuncture Education in Classic Non-superior Schools.

Institute	Website	Acupuncture Courses in the TCM Program	Acupuncture Courses in the Continuing Training Program	Connection with TCM Universities in China
ESMTC established in 1992	https://esmtc.pt (accessed on 20 October 2022)	Topography of meridians and points I, II, III; basics of clinical acupuncture; introduction to acupuncture and moxibustion; clinical acupuncture I, II; techniques of clinical acupuncture.	No	Nanjing University of Chinese Medicine
UMC established in 1997	https://umc.pt (accessed on 20 October 2022)	Meridians and points of acupuncture I, II; acupuncture, moxibustion and cupping therapies I, II, III, IV; somatopuncture and ear acupuncture (Online only).	1. Aesthetic acupuncture; 2. Ear acupuncture; 3. Shun Fa’s Scalp acupuncture; 4. Complementary practice of acupuncture and tuina; 5. Introduction to practice of Chromopuncture	Chengdu University of Traditional Chinese Medicine
IPN established in 1999	https://www.ipnaturologia.com (accessed on 20 October 2022)	Meridian structure and Point location I, II; techniques of acupuncture and moxibustion I, II; auriculotherapy I, II; practice of auriculotherapy.	1. Electroacupuncture; 2. Auriculotherapy	Beijing University of Chinese Medicine

**Table 5 healthcare-11-01389-t005:** Post-graduate Acupuncture Teaching Plans in Portugal Universities.

University	Duration	Content	Proportion of Traditional Theories and Practice
University of Porto since 2003	30 ETCS (810 h)	Traditional theories and practice in chief with a little modern medicine	About 90%
University of Coimbra since 2007	120 ETCS (3240 h)	Modern medicine in chief with a little traditional theories and practice	About 20%
Nova University Lisbon since 2010	12 ETCS (324 h)	All in modern medicine	0%
University of Minho since 2012	12 ETCS (316 h)	All in modern medicine	0%

## Data Availability

Not applicable.

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
