# Peer review of "Insights into Current Education of Acupuncture as a Non-Conventional Therapy in Portugal"

_healthcare, 2023, doi:10.3390/healthcare11101389_

Round 1

Reviewer 1 Report

Dear authors, 

This manuscript outlines the non-conventional therapies of acupuncture in Portugal. It can contribute the future development and improvement of acupuncture in Portugal as well as in other countries that accept acupuncture as a non-conventional therapy for diseases.

In general, this manuscript has been well written, with sufficient information regarding acupuncture. However, this manuscript requires improvements especially in terms of methodologies and results. The following are the specific comments and suggestions:

1.      Abstract:

Please add a brief introduction regarding non-conventional therapy (acupuncture)

Please add the aim of this study after the introduction of acupuncture. 

Line 21: Please add the type of questionnaire used during the investigation. Specifically, please state the number of people involved in the investigation (sample size).

Line 22:” …Portugal, we showed….” For this sentence, better to break into two sentences, to differentiate between methodology and result.

2.      Introduction:

Line 36: Please define acupuncture and state the common diseases that can be treated by acupuncture.

Line 56- 64:

“… flawed in some respects. Therefore, the aim of this study was to determine [add this] the current acupuncture education in Portugal, expose problems regarding the legislation, teaching contents, staff, research, internship, etc., and then it is possible to provide constructive suggestions, which could be very meaningful and thought-provoking to the future development of acupuncture in Portugal, in Portuguese-speaking countries, and in other countries that welcome acupuncture and intend to have better legislation and application, even for other non- conventional therapies.  [In addition, I think this sentence is too long. Please rephrase and shorten it accordingly].

Line 56-58: Please put this sentence “Through investigation ……acupuncture education in Portugal” into Methodology section.

3.      Methodology

Line 64: Please make sure that methodology part is exist in this research article, which is an important part. There should be an explanation on the sample size, what kind of questionnaire was given to the respondents.

4.      Results 

Please make sure that result part is exist in this research article, which is also an important part. I think that the lines between 65-210 are the results of this article. Therefore, please put a subtopic “Results” before the line 65.

Discussion: Line 366: Please add limitations and future perspectives of this study. Alternatively, please make it as a new subtopic (5. Limitations and Future perspectives)

I think this article needs moderate editing of English language, as I could see that several sentences are quite complicated and too long. Please rephrase the long sentences into a simpler and easier to understand.

Author Response

Thank you very much for processing and reviewing our manuscript entitled "Insights into Current Education of Acupuncture as a Non-conventional Therapy in Portugal”. Your valuable comments and instructions are highly appreciated by us. The MS has been revised substantially and the revisions are all marked using the “Track Changes” function. The English is further polished and many long sentences have been broken into shorter ones for easier understanding. Once again, we highly appreciate your time and efforts. We believe that our MS has been considerably improved because of your valuable comments and instructions. Point by point responses to the comments are as follows

  1. Abstract:

Please add a brief introduction regarding non-conventional therapy (acupuncture)

Please add the aim of this study after the introduction of acupuncture.

Line 21: Please add the type of questionnaire used during the investigation. Specifically, please state the number of people involved in the investigation (sample size).

Line 22:” …Portugal, we showed….” For this sentence, better to break into two sentences, to differentiate between methodology and result.

Response: Thank you for your valuable comments and instructions. 

In the abstract, a brief introduction to acupuncture was added: “Acupuncture, being an ancient practice for healthcare in China, is now widely used in the world and regarded as a non-conventional therapy (NCT) in many Western countries.”

The aim of the study was added: “This article aims to disclose the current education of acupuncture as a NCT in Portugal…”.

    Two tables were inserted with questions made during the interviews and a flowdiagram of sample size were added.

” …Portugal, we showed….” has been broken into two sentences.

  1. Introduction:

Line 36: Please define acupuncture and state the common diseases that can be treated by acupuncture.

Line 56- 64:“… flawed in some respects. Therefore, the aim of this study was to determine [add this] the current acupuncture education in Portugal, expose problems regarding the legislation, teaching contents, staff, research, internship, etc., and then it is possible to provide constructive suggestions, which could be very meaningful and thought-provoking to the future development of acupuncture in Portugal, in Portuguese-speaking countries, and in other countries that welcome acupuncture and intend to have better legislation and application, even for other non- conventional therapies.  [In addition, I think this sentence is too long. Please rephrase and shorten it accordingly].

Line 56-58: Please put this sentence “Through investigation ……acupuncture education in Portugal” into Methodology section.

Response: Thank you for your valuable comments and instructions.

    In the Introduction, the definition of acupuncture and the common diseases that can be treated are added in Line 37-71: “Acupuncture is a technique in which practitioners insert fine needles into the skin to treat health problems. It originated from traditional Chinese medicine at least 2,500 years ago, but has gained popularity worldwide since the 1970s.”; “WHO recognizes 28 diseases, symptoms, or conditions for which acupuncture has been shown as an effective form of treatment., and it also recognizes acupuncture’s therapeutic effects for over 55 diseases, symptoms, or conditions. A recent overview of acupuncture systematic reviews found that of 77 diseases investigated, acupuncture showed a moderate or large effect with moderate or high certainty evidence in eight diseases or conditions, namely post-stroke aphasia, neck and shoulder pain, myofascial pain, fibromyalgia related pain, non-specific lower back pain, lactation success rate within 24 hours of delivery, vascular dementia symptoms, and allergic rhinitis nasal symptoms.”

Previous Line 56- 64 have been rephrased and “the aim of this study was to determine” has been added.

The sentence “Through investigation ……acupuncture education in Portugal” has been put into the Methodology section.

  1. Methodology

Line 64: Please make sure that methodology part is exist in this research article, which is an important part. There should be an explanation on the sample size, what kind of questionnaire was given to the respondents.

Response: Thank you once again, you are totally right. Now we mentioned the methodology chapter and inserted 2 tables with the questions that were made during the interviews and a flow diagram of sample size with inclusion/exclusion process.

  1. Results

Please make sure that result part is exist in this research article, which is also an important part. I think that the lines between 65-210 are the results of this article. Therefore, please put a subtopic “Results” before the line 65.

Response: Thank you for your valuable comments and instructions. You are right that the “Results” are the information of the acupuncture education in Portugal and many problems in the discussion part. Subtitle “Results” has been added in the article and it becomes more structured.

  1. 5. Line 366: Please add limitations and future perspectives of this study. Alternatively, please make it as a new subtopic (5. Limitations and Future perspectives)

Response: Thank you for your valuable comments and instructions. The part “5. Limitations and Future Perspectives” was added in Line 693-699: “Due to the limitations of time, funds and communication, only a minority of institutions providing acupuncture education had been visited and a few relevant people had been interviewed. The data collected would thus not be comprehensive enough, and some underlining problems might haven’t been revealed. More field work and interviews could be further carried out to discover more detailed picture of the acupuncture education in Portugal.”

Once again, your time and efforts are highly and sincerely appreciated by us. Your valuable comments and instructions help us improve our MS substantially. We sincerely hope that our MS can be accepted and published in your influential journal. Have a good day!

Reviewer 2 Report

The topic is original with sufficient references, it has addressed a specific gap in the field, not only for Portugal, but also other Western country, about TCM education outside of China.

It would be better if update the table with all the important figures of all institutions, such length of course, number of students and stuff in which number of fulltime stuff, degree of award, learning hours, any teaching clinic, etc.

Page 6 Line 291"In China, high school graduates with a background in science or art can all apply for a TCM degree education," needs to clarify, as previous knowledge was not like this. TCM education in China should have background of science but not art. It might be in some universities but not majority. 

Author Response

Dear editor and reviewers concerned:

Thank you very much for processing and reviewing our manuscript entitled "Insights into Current Education of Acupuncture as a Non-conventional Therapy in Portugal”. Your valuable comments and instructions are highly appreciated by us. The MS has been revised substantially and the revisions are all marked using the “Track Changes” function. The English is further polished and many long sentences have been broken into shorter ones for easier understanding. Once again, we highly appreciate your time and efforts. We believe that our MS has been considerably improved because of your valuable comments and instructions. Point by point responses to the comments are as follows:

Comments to the Submitting Author

  1. It would be better if update the table with all the important figures of all institutions, such length of course, number of students and stuff in which number of fulltime stuff, degree of award, learning hours, any teaching clinic, etc.

Response: Thank you for your valuable comments and instructions. Table 3 in Line 342-361 is added to show some more information about acupuncture degree education in all polytechnic schools. However, some information is still missing, because the data could not be available online or the person we interviewed may not know the specific data. More comprehensive survey would be required in the future.

  1. Page 6 Line 291"In China, high school graduates with a background in science or art can all apply for a TCM degree education," needs to clarify, as previous knowledge was not like this. TCM education in China should have background of science but not art. It might be in some universities but not majority. 

Response: Thank you for your valuable comments and instructions. In the time when the first author of the article was a TCM student in the university in China, high school graduates with a background in science or art could all be enrolled. However, after the first author had a careful check, it shows you are right. At present, the TCM degree education in China mostly only allow students with a background of science, with very few vacancies for those with a background of art. We thought this sentence was not important, and to prevent confusing, we deleted this sentence in the article

Once again, your time and efforts are highly and sincerely appreciated by us. Your valuable comments and instructions help us improve our MS substantially. We sincerely hope that our MS can be accepted and published in your influential journal. Have a good day!

Reviewer 3 Report

The authors should be commended for having the courage to write this important paper on a very difficult and controversial subject.

However, for a reader with little knowledge or understanding of legal processes, it is quite difficult to follow, at least at the start of the paper. A clearer structure would help, and, if possible, some explanation in simple language of what the various laws state and imply.

These laws simply mention acupuncture, without examining what potential results the legislation might have. They pose a real threat to the continued practice in Portugal of acupuncture as anything more than a 'technique' like dry needling.   

It would be useful to provide more background information on the situation in other countries (in the EU and elsewhere), and the contradictions and similarities between 'medical' and 'traditional' acupuncture. 

Table 1 is poorly presented and somewhat confusing. 

Summarising the Discussion in bullet points might be  helpful.

The paper is off-putting for a native English speaker. The language is sometimes ponderous. It could be simplified and the findings presented in a more inviting way. 

However, I do appreciate that this is not a simple, everyday subject to write about, and that the authors have had to tread carefully in their presentation.

Author Response

Dear editor and reviewers concerned:

Thank you very much for processing and reviewing our manuscript entitled "Insights into Current Education of Acupuncture as a Non-conventional Therapy in Portugal”. Your valuable comments and instructions are highly appreciated by us. The MS has been revised substantially and the revisions are all marked using the “Track Changes” function. The English is further polished and many long sentences have been broken into shorter ones for easier understanding. Once again, we highly appreciate your time and efforts. We believe that our MS has been considerably improved because of your valuable comments and instructions. Point by point responses to the comments are as follows

Comments to the Submitting Author

  1. However, for a reader with little knowledge or understanding of legal processes, it is quite difficult to follow, at least at the start of the paper. A clearer structure would help, and, if possible, some explanation in simple language of what the various laws state and imply.

Response: Thank you for your valuable comments and instructions. At the start of the paper in the third paragraph, we inserted the first law of acupuncture in detail with explanations for better understanding in Line 81-89. All the laws in the “Major Policies” had been expressed in an easier language and implications were given.

  1. These laws simply mention acupuncture, without examining what potential results the legislation might have. They pose a real threat to the continued practice in Portugal of acupuncture as anything more than a 'technique' like dry needling.

Response: Thank you for your valuable comments and instructions. Acupuncture mentioned in the “Major Policies” is exclusively based on traditional Chinese medicine as a non-conventional therapy rather than dry needling or medical acupuncture. As for this, we added a law that defines the acupuncture in this part for better understanding in Line 217-222: On October 8th, 2014, acupuncture as a non-conventional therapy was defined in Ordinance 207-F/2014, stating that “Acupuncture is a therapy that uses methods of diagnosis, prescription and treatment based on axioms and theories of acupuncture meridians, points and reflex zones of the human body to prevent and treat energetic, physical and psychic disharmonies.” It also clearly states that the theory and practice of acupuncture are based on traditional Chinese medicine (TCM).

    Therefore, the acupuncture as a non-conventional therapy in our article solely refers to the acupuncture based on traditional Chinese medicine.

  1. It would be useful to provide more background information on the situation in other countries (in the EU and elsewhere), and the contradictions and similarities between 'medical' and 'traditional' acupuncture.

Response: Thank you for your valuable comments and instructions. The background information on the acupuncture situation in the EU was added in the introduction at the beginning of the second paragraph Line 72-74: In the West, acupuncture as a distinct therapeutic system is recognised by law in 12 European Union (EU) Member States, and the medical association/ council/ chamber in many countries has recognised acupuncture as an additional medical qualification.

The comparison between medical acupuncture and traditional acupuncture (non-conventional therapy) was scattered in the article, such as Line 232-233 “(traditional acupuncture) Its curricular content … includes basic and clinical subjects of modern medicine, basic theories of TCM…”; Line 451 “medical acupuncture, … is exclusively learned and practiced by medical students and doctors”; Line 485-487 “teaching acupuncture to medical doctors is best achieved by teaching western style acupuncture, focusing on the needling technique, and basing the treatment plans in sound knowledge of anatomy and physiology”. However, the contradictions and similarities between 'medical' and 'traditional' acupuncture are generally the same in other countries in the West, while this article mainly focuses on the acupuncture education as a non-conventional therapy in Portugal, so such a comparison is not conducted intensively and extensively.

  1. Table 1 is poorly presented and somewhat confusing.

Response: Thank you for your valuable comments and instructions. The headings of Table 1 (now Table 4) have been revised and we hope it could be easier for understanding.

  1. Summarising the Discussion in bullet points might be helpful.

Response: It’s a very good suggestion. We have revised the discussion part in bullet points. It really becomes more academic and well structured.

Once again, your time and efforts are highly and sincerely appreciated by us. Your valuable comments and instructions help us improve our MS substantially. We sincerely hope that our MS can be accepted and published in your influential journal. Have a good day!

Round 2

Reviewer 1 Report

Dear Authors,

Thank you for revising the manuscript accordingly. I think this manuscript has a better structure for a research article. Well done!

However, there is a simple correction that needed to be done:

1. line 43: ....194 ones. [2]

for this, it should be written ...194 ones [2]. The full stop should be put after the reference number.

Please check thoroughly for the whole manuscript: line 45,line 51,line 54, line 58, line 60, line 67, line 120,line 138, line 155.

2. Line 241: ......University of Minho.[22][23]

Please write .....University of Minho [22, 23]. Please make sure to use comma if more than one reference is used.

3. Line 40: no full stop : 1970s [1] According....

Author Response

My dear reviewer, thank you so much for your meticulous checkup and constructive suggestions. Our article has been improved greatly and small mistakes have also been corrected under your guidance.